# De novo cholesterol biosynthesis in bacteria

Alysha K. Lee[1], Jeremy H. Wei[1] & Paula V. Welander ⓘ [1] ✉

Eukaryotes produce highly modified sterols, including cholesterol, essential to eukaryotic physiology. Although few bacterial species are known to produce sterols, de novo production of cholesterol or other complex sterols in bacteria has not been reported. Here, we show that the marine myxobacterium *Enhygromyxa salina* produces cholesterol and provide evidence for further downstream modifications. Through bioinformatic analysis we identify a putative cholesterol biosynthesis pathway in *E. salina* largely homologous to the eukaryotic pathway. However, experimental evidence indicates that complete demethylation at C-4 occurs through unique bacterial proteins, distinguishing bacterial and eukaryotic cholesterol biosynthesis. Additionally, proteins from the cyanobacterium *Calothrix* sp. NIES-4105 are also capable of fully demethylating sterols at the C-4 position, suggesting complex sterol biosynthesis may be found in other bacterial phyla. Our results reveal an unappreciated complexity in bacterial sterol production that rivals eukaryotes and highlight the complicated evolutionary relationship between sterol biosynthesis in the bacterial and eukaryotic domains.

Sterols are a class of ubiquitous and essential eukaryotic lipids important in a variety of physiological functions including cell signaling, membrane homeostasis, and developmental timing[1–3]. Eukaryotic sterol biosynthesis has been studied extensively revealing complex biosynthetic pathways including a shared set of enzymes used to produce similar sterol end products that vary in the level of unsaturations, demethylations, and alkalytions[4,5]. The order of reactions in these biosynthetic pathways dictate the production and accumulation of sterol intermediates which play a regulatory role in lipid homeostasis[6,7], downstream product biosynthesis[8], and stress response[9]. The chemical modifications required to synthesize cholesterol in vertebrates, phytosterols in plants, and ergosterol in fungi are essential to the biophysical properties of these lipids, effecting localization and membrane dynamics in their respective organisms[10–12]. These sterols also serve as a branching point for downstream metabolite biosynthesis. Oxidation reactions are involved in converting sterols into a wide range of compounds including oxysterols, bile acids, steroid hormones, and brassinosteroids which can all function as ligands in various signaling pathways[13–15]. Eukaryotes also conjugate sterols to sugars, proteins, and other lipids, further expanding the function of sterols to include cell defense, energy storage, digestion, and signaling[16–18]. Overall, the complexity of eukaryotic sterol biosynthesis is reflective of the diverse functions and significant roles these lipids play in eukaryotic physiology.

While sterol biosynthesis and function has been well-studied in eukaryotic organisms, bacterial sterol synthesis and function is comparatively underexplored. Several bacteria, including aerobic methanotrophs, Planctomycetes, and various myxobacteria are known to produce sterols de novo[19]. Unlike eukaryotes, these bacteria largely produce lanosterol, parkeol, or cycloartenol, the initial cyclization products of oxidosqualene cyclase (OSC)[20–22]. However, some bacteria perform additional chemical modifications during sterol synthesis; methanotrophic bacteria modify sterols into distinct monomethylated structures specific to Methylococcaceae[19], sterol-producing Planctomycetes conjugate sterols to an unidentified macromolecule[21], and several Myxococcota[23] produce intermediates in the cholesterol biosynthesis pathway, including zymosterol[19,22,24]. Furthermore, phylogenomic studies have expanded the number of potential bacterial sterol producers, identifying the genes required for both cyclization and downstream modifications in phyla across the bacterial domain[25,26]. Several of these bacteria harbor the genetic potential to produce the biosynthetically complex sterols associated with eukaryotes, including cholesterol, however lipid analyses have yet to confirm the presence of these sterols in bacteria[5].

[1]Department of Earth System Science, Stanford University, Stanford, CA 94305, USA. ✉e-mail: welander@stanford.edu

The discrepancy between the genomic capacity for complex sterol production and the observed sterols in bacteria prompted us to undertake more comprehensive lipid analyses of sterol-producing bacteria. Previous phylogenetic studies and our own initial analysis of sterol biosynthesis genes of the marine myxobacteria *Enhygromyxa salina* suggested the potential for production of more biosynthetically complex sterols than we have previously observed. We hypothesized previous analyses may have underestimated the bacterial sterol inventory due to limited biomass and/or inadequate lipid extraction techniques, motivating us to reanalyze sterols in this bacterium. Similarly, analysis of the cyanobacteria *Calothrix* sp. NIES-4105 genome identified a cluster of putative sterol biosynthesis genes including those required for complex sterol biosynthesis, leading us to expand our sterol analysis to this bacterium.

In this work, we revisit sterol analysis in *E. salina*, revealing a capacity to synthesize cholesterol. Our bioinformatic analysis identified homologs to some, but not all, of the steps in eukaryotic cholesterol biosynthesis, differentiating bacterial cholesterol production from that in eukaryotes. Further, we demonstrate that complete demethylation at C-4, an essential step in eukaryotic sterol biosynthesis, occurs through distinct bacterial proteins in *E. salina* and that homologs found in the phylogenetically distinct sterol producing bacterium, *Calothrix*, function similarly. Altogether, our analyses of sterol biosynthesis in these phylogenetically distinct bacteria suggest a complex biology underpinning bacterial cholesterol production and raise broader questions about sterol biosynthesis, evolution, and function.

## Results

### *Enhygromyxa salina* produces free and conjugated sterols

We previously demonstrated the production of zymosterol, a biosynthetic intermediate in cholesterol synthesis, in lipid extracts of the myxobacterium *Enhygromyxa salina*[19]. However, *E. salina* is a social predatory bacterium often cultured on solid agar containing whole cell yeast, limiting the amount of biomass available for extensive lipid analyses[27]. These culturing conditions also present a potential source of sterol contamination as both agar and supplemental yeast may contain sterols. To better assess the sterol inventory of this organism, we grew *E. salina* in a liquid medium supplemented with whole cell *Escherichia coli*, which does not natively produce sterols. These culturing conditions increased extractable lipids 50-fold, allowing for more extensive analyses.

We first extracted free lipids from *E. salina* biomass through a Bligh-Dyer extraction[28]. These initial analyses revealed a variety of sterols, including cholesterol (Fig. 1a; Supplementary Fig. 1, Supplementary Fig. 2). We also detected 4,4-dimethylcholesta-8,24-dienol, an intermediate only demethylated at C-14, indicating *E. salina* performs C-14 demethylation before C-4 demethylation, like fungi and animals, and not after removal of the first C-4 methyl group, like plants[5]. We also detected several downstream intermediates unsaturated at C-24 including zymosterol, cholesta-7,24-dienol, and desmosterol but no intermediates saturated at C-24. The accumulation of only C-24 unsaturated intermediates suggests *E. salina* favors the Bloch cholesterol biosynthesis pathway, where C-24 reduction occurs as the final step in cholesterol biosynthesis (Supplementary Fig. 3)[29]. Quantification of cholesterol as well as the intermediates desmosterol and zymosterol revealed intermediates to be at concentrations similar to or greater than cholesterol (Supplementary Table 1). Finally, to control for potential contamination, we extracted both culturing medium and concentrated supplemental *E. coli* without bacterial biomass. Both the media and extraction controls were devoid of sterols (Supplementary Fig. 4).

To assess *E. salina* for potential sterol conjugates, we hydrolyzed both total lipid extract (TLE) and cell biomass with either methanolic base, which cleaves ester bound lipids, or methanolic acid, which cleaves ester and ether bound lipids[30]. Hydrolysis of TLE with methanolic base did not release additional sterols but hydrolysis with methanolic acid released 25-hydroxycholesterol (25-OHC) as well as additional oxysterols and other potential modified sterol compounds (Fig. 1b and Supplementary Fig. 5). The presence of oxysterols after

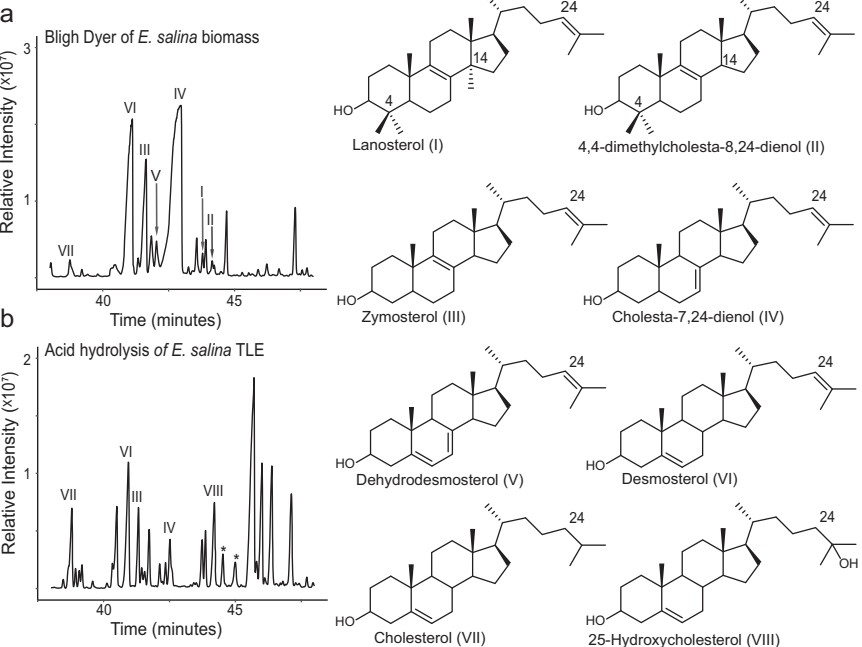

**Fig. 1 | *E. salina* synthesizes unbound and bound sterols. a** Total ion chromatogram of free sterols from *E. salina* extracted using modified Bligh Dyer procedure. Identified sterols included cholesterol (VII) as well as C-24 unsaturated intermediates (I-VI). **b** Total ion chromatogram of ether and ester bound sterols released from *E. salina* lipid extracts by acid hydrolysis. Hydrolysis released additional sterols, including 25-hydroxycholesterol (VIII) and other putative hydroxysterols (denoted by asterisk). All lipids were derivatized to trimethylsilyl groups. Mass spectra of identified sterols are shown in Supplementary Fig. 1. Source data are provided as a Source Data file.

acid hydrolysis of TLE, but not after base hydrolysis, suggests that these conjugations are mediated by an ether bond. To further characterize the chemical nature of sterol conjugates, we separated extractable lipids by polarity using Si-gel column chromatography and hydrolyzed fractions with methanolic acid. After hydrolysis, all sterols, including hydroxysterols, were only present in the alcohol fraction, suggesting conjugated sterols are similar in polarity to free sterols. Direct hydrolysis of cell biomass with either methanolic base or acid did not alter the *E. salina* sterol profile from that of the hydrolyzed TLEs. To confirm these additional sterols were not the products of degradation or autooxidation during hydrolysis, we first acid hydrolyzed cholesterol and desmosterol standards and did not detect production of any hydroxysterols. We also added butylated hydroxytoluene (BHT) to an *E. salina* acid hydrolysis extraction and still detected 25-hydroxycholesterol and the other potential modified sterols observed in extractions without BHT (Supplementary Fig. 6).

## *E. salina* harbors eukaryotic cholesterol biosynthesis protein homologs

Production of cholesterol by *E. salina* prompted us to search for a putative biosynthesis pathway in its genome. We conducted a BLASTp search (<$1 \times e^{-30}$, 30% ID) against sterol biosynthesis proteins from eukaryotes and bacteria[4,31]. We chose a restrictive cut-off for these BLASTp searches to better ensure identified proteins are likely involved in sterol biosynthesis, as proteins in the cholesterol biosynthesis pathway belong to large superfamilies that include functions outside sterol biosynthesis[5]. We detected homologs for nearly every step in the eukaryotic cholesterol biosynthesis pathway in the *E. salina* genome (Fig. 2 and Supplementary Table 2), although further biochemical characterization is needed to demonstrate the identified proteins do not perform additional reactions in the cell. Notably, sterol biosynthesis genes are not localized in gene clusters, as observed in other sterol-producing bacteria[32], but instead found in separate loci across the genome (Supplementary Fig. 7). Using this newly constructed cholesterol biosynthesis pathway, we next searched for homologs in other bacteria with demonstrated genomic capacity for sterol production (<$1 \times e^{-50}$, 30% ID). We identified 103 bacterial isolates and 21 metagenome assembled genomes with squalene monooxygenase (SMO) and oxidosqualene cyclase (OSC), the first committed steps in sterol biosynthesis. Of these OSC and SMO containing bacterial genomes and metagenomes, only the other sequenced isolates of *E. salina* share the required homologs for the eukaryotic cholesterol biosynthesis pathway, indicating cholesterol production is likely a shared feature of these *E. salina* species (Supplementary Data 1).

## C-4 demethylation occurs through distinct bacterial proteins

While *E. salina* shares much of its cholesterol biosynthesis pathway with eukaryotes, it does not have homologs to the three eukaryotic proteins responsible for C-4 demethylation. Instead, *E. salina* has homologs to the dioxygenase-reductase pair, SdmAB, used by aerobic methanotrophs to remove only a single methyl group at C-4[31]. To test whether the *E. salina* SdmAB homologs are sufficient to remove both methyl groups at C-4, as is required to produce cholesterol, we employed a heterologous expression system in an *E. coli* strain engineered to overproduce the substrate lanosterol (Fig. 3a)[33]. Expression of the SdmA homolog alone resulted in production of a 4-methylaldehyde intermediate. Expression of the SdmB homolog alone did not generate any demethylation intermediates (Supplementary Figs. 8 and 9). Interestingly, co-expression of the SdmAB homologs did not result in removal of any methyl groups at C-4 (Fig. 3b), indicating these two proteins are not sufficient to fully demethylate lanosterol at the C-4 position in *E. salina* and distinguishing SdmAB in *E. salina* from the homologs found in aerobic methanotrophs.

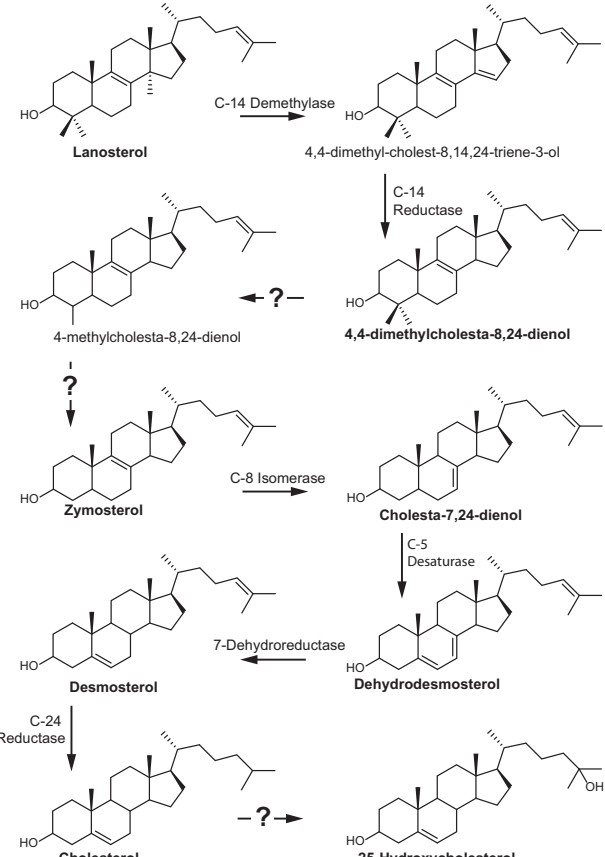

**Fig. 2 | A putative cholesterol biosynthesis pathway in *E. salina*.** Homologs to the canonical eukaryotic proteins for cholesterol biosynthesis were identified by BLASTp search (<$1 \times e^{-30}$, 30% ID; Supplementary Table 2). Sterols identified in *E. salina* biomass are bolded. *E. salina* does not have homologs to the eukaryotic enzymes responsible for C-4 demethylation and C-25 hydroxylation. Bacterial enzymes for complete C-4 sterol demethylation and C-25 hydroxylation have not been identified and are denoted by a question mark. Genomic context for identified *E. salina* genes are shown in Supplementary Fig. 7. Source data are provided as a Source Data file.

We next sought to determine whether the lack of C-4 demethylation by the *E. salina* homologs was tied to SdmA or SdmB. To do so, we leveraged the SdmA and SdmB homologs from *Methylococcus capsulatus* - which together remove a single methyl group in our expression system - and expressed them with their reciprocal SdmA or SdmB partner from *E. salina*. Expression of SdmA from *E. salina* with SdmB from *M. capsulatus* resulted in removal of a single methyl group at C-4, while expression of SdmB from *E. salina* with SdmA from *M. capsulatus* resulted in no demethylation at C-4 (Supplementary Fig. 10). Thus, we propose the *E. salina* SdmA homolog can carry out the oxygenation reactions required to demethylate, but the *E. salina* SdmB homolog is insufficient to carry out both the decarboxylation and reduction reactions required for demethylation at the C-4 position[31]. These results led us to reassess *E. salina* for enzymes with the potential to carry out the decarboxylation and reduction reactions.

Through an additional BLASTp search of the *E. salina* genome, we identified another SDR-type reductase homologous to SdmB, SdmC ($1 \times e^{-120}$, 51% identity), restricted to the Myxococcota suborder *Nannocystaceae* (Supplementary Fig. 11). Expression of SdmC alone did not result in accumulation of any demethylation intermediates (Supplementary Fig. 8). However, co-expression of SdmC with SdmA from *E. salina* resulted in removal of a single methyl group at C-4, suggesting SdmC is capable of decarboxylating the oxidized methyl group and

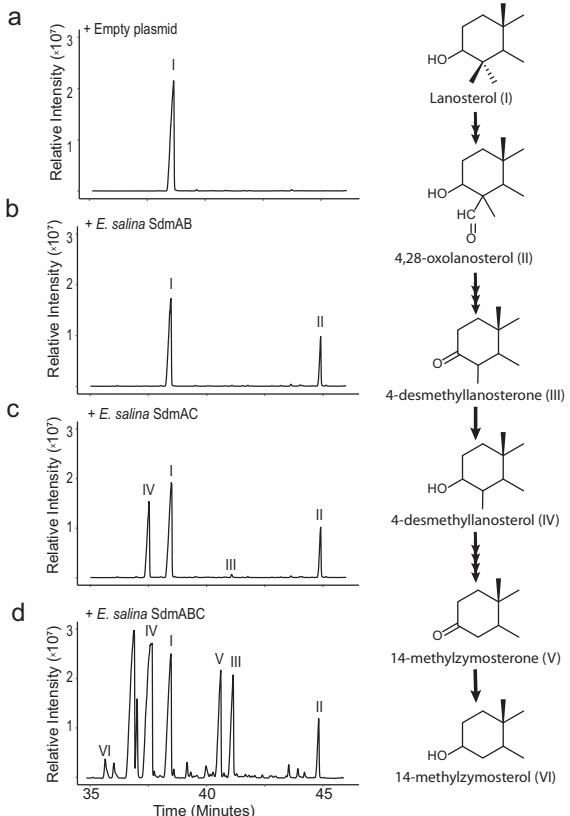

**Fig. 3 | C-4 demethylation in *E. salina*. a** Total ion chromatograms showing lanosterol (I) substrate production in our *E. coli* heterologous expression system. **b** Co-expression of SdmAB homologs from *E. salina*, resulting in production of only a 4-methylaldehyde intermediate (III). These two enzymes are insufficient to demethylate at C-4, distinguishing them from SdmAB homologs in aerobic methanotrophs. **c** Co-expression of SdmAC homologs from *E. salina*, resulting in removal of single methyl group at C-4. **d** Co-expression of SdmABC homologs from *E. salina*, resulting complete demethylation at C-4. Lipids were derivatized to tri-methylsilyl groups. C-4 demethylation intermediates were confirmed in a previous study and identified here by comparison to published mass spectra (32). Mass spectra of identified sterols are shown in Supplementary Fig. 9. Source data are provided as a Source Data file.

reducing the remaining ketone at C-3 into a hydroxyl as was shown for the *M. capsulatus* SdmB (Fig. 3c)[31]. Additionally, co-expression of SdmC with *E. salina* SdmAB results in removal of both methyl groups at C-4 and production of additional demethylation intermediates (Fig. 3d). Thus, full demethylation at the C-4 position in *E. salina* is distinct from both bacterial C-4 demethylation in aerobic methanotrophs and eukaryotes (Supplementary Fig. 12).

While SdmC appears restricted to a specific myxobacterial sub-order, SdmAB homologs can be found in a variety of isolate and metagenome acquired genomes (MAGs) throughout the domain. This includes 31 bacteria across five different phyla (Supplementary Data 1). We were interested in determining if these other SdmAB homologs were also sufficient to double demethylate lanosterol or if they functioned more similarly to the SdmAB homologs in aerobic methanotrophs, and only remove a single methyl group at C-4. In particular, the cyanobacterium *Calothrix* sp. NIES-4105 genome harbors SdmAB homologs in a 21 kb gene cluster, which also contains homologs for oxidosqualene production, cyclization, C-14 demethylation, C-8 isomerization and several additional proteins likely involved in sterol biosynthesis (Fig. 4a and Supplementary Table 2). This diversity of sterol biosynthesis genes in a cyanobacterium has not been previously observed suggesting that this *Calothrix* strain could produce complex

sterols that are fully demethylated at the C-4 position. To test this, we first heterologously expressed the *Calothrix osc* homolog in an oxidosqualene-producing *E. coli* strain resulting in lanosterol production and confirming that the *Calothrix* cyclase is functional (Supplementary Fig. 13). We next expressed the *Calothrix* SdmAB homologs to determine if these proteins were capable of demethylation at C-4. Expression of the SdmA homolog alone resulted in production of a 4-methylaldehyde intermediate while expression of the SdmB homolog alone resulted in the production of lanosterone (Supplementary Figs. 8 and 9). Co-expression of *Calothrix* SdmAB was sufficient to remove both methyl groups at C-4, demonstrating a third bacterial C-4 demethylation pathway, distinct from *E. salina*, aerobic methanotrophs, and eukaryotes (Fig. 4b and Supplementary Fig. 12).

### *Calothrix* sp. NIES-4105 may produce biosynthetically complex sterols
Given the genetic and biochemical evidence for complex sterol production in *Calothrix*, we next analyzed sterols produced by this bacterium. In our initial analysis, we were only able to detect sterols after acid hydrolyzing cell biomass (Supplementary Figs. 1, 2, and 13). These sterols included cholesterol as well as both C-24 unsaturated and C-24 saturated intermediates. We also identified 25-OHC in hydrolyzed *Calothrix* extracts, further suggesting sterols maybe conjugated in *Calothrix*. However, throughout the course of lipid analyses, *Calothrix* ceased all sterol production. To determine if loss of sterol production in *Calothrix* was due to a genetic mutation in the sterol biosynthetic gene cluster, we sequenced the genome of the serial passaged strain and compared it to the reference genome of *Calothrix* sp. NIES-4105. We detected 14 mutations the genome of the serial passaged strain compared to the reference genome available in the JGI IMG and NCBI databases, however, none of these mutations occurred in or upstream of the sterol biosynthesis gene cluster (Supplementary Table 3). While these results render our initial sterol analysis of *Calothrix* inconclusive, our findings do suggest cyanobacteria as a potential source of biosynthetically complex sterols and a phylum to further consider when investigating bacterial sterol biosynthesis.

### Discussion
In this study, we demonstrate bacteria harbor the capacity to synthesize cholesterol de novo, displaying a biosynthetic complexity often associated with eukaryotes and suggesting the possibility for nuanced physiological functions for sterols in bacteria. Our analysis of *E. salina* identified cholesterol as well as intermediates in the cholesterol biosynthesis pathway. Quantification of these lipids shows that biosynthetic intermediates occur at greater or similar concentrations to cholesterol itself. This is distinct from cholesterol production in many eukaryotes, where intermediates may be present but often at concentrations orders of magnitude lower than cholesterol or other 'end-product' sterols[34,35]. Given the high concentrations of these intermediates in this bacterium relative to cholesterol, we posit that cholesterol intermediates, such as desmosterol and zymosterol, serve as functional lipids themselves, though what those functions are and if they differ from cholesterol, remains unclear.

Our analyses of *Calothrix* provided genomic and biochemical evidence of a capacity for complex sterol production, however, sterol production by this bacterium remains uncertain. Historically, de novo sterol biosynthesis in cyanobacteria has been controversial. Early sterol analysis of cyanobacteria suggested a capacity for complex sterol biosynthesis[36]. However there was a lack of supporting genomic evidence and, in several cases, sterol production was later demonstrated to be caused by fungal contamination[37,38]. Our work demonstrating functionality of sterol biosynthesis proteins from *Calothrix*, combined with bioinformatic data showing a similar genetic capacity in other cyanobacteria encourages future investigation of the biosynthetic enzymes found in these bacteria and further analysis of the

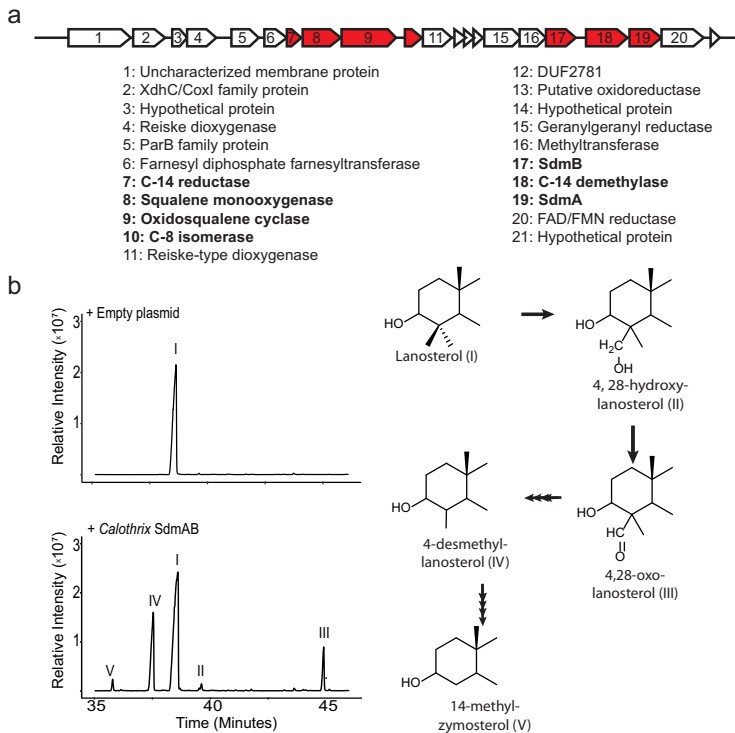

**Fig. 4 | C-4 demethylation in *Calothrix* sp. NIES-4105. a** Sterol biosynthesis genes in *Calothrix* are localized in a single gene cluster. Genes identified by our BLAST search are indicated in red and text labels bolded. Of note are several other genes annotated as putative biosynthesis genes which may be responsible for carrying out additional steps in cholesterol biosynthesis. **b** Total ion chromatograms of lanosterol substrate production in our heterologous expression system and coexpression of SdmAB homologs from *Calothrix* resulting in complete demethylation at C-4. Lipids were derivatized to trimethylsilyl groups. C-4 demethylation intermediates were confirmed in a previous study and identified here by comparison to published mass spectra (32). Mass spectra of identified sterols are shown in Supplementary Fig. 9. Source data are provided as a Source Data file.

sterols they produce. Additionally, we identified the genes required for complex sterol biosynthesis, including C-14 and C-4 demethylation, in a diverse set of bacteria including other myxobacteria and cyanobacteria, as well as actinobacteria, acidobacteria, and nitrospirea (Supplementary Data 1). In many of these bacteria, sterol production has yet to be analyzed. Coupling more robust lipid analysis with further study of bacterial sterol biosynthesis would allow us to better grasp the distribution, diversity, and complexity of sterols in bacteria.

The presence of free and conjugated sterols in *E. salina* provides another example of the biosynthetic complexity of bacterial sterol production. Bacterial sterol conjugation is not limited to *E. salina*; other bacteria conjugate sterols, either as the product of de novo biosynthesis[21] or modification of exogenously acquired sterols[39,40]. These different sterol pools suggest the potential for a varied set of physiological roles for sterols in bacteria. Conjugations to other macromolecules impact the biophysical properties of sterols[41] and, in eukaryotic systems, these modified lipids are involved in specific functions including lipid storage and cell defense[16,17]. While intact sterol conjugates have not been identified in sterol-producing bacteria, the sterol conjugates identified in eukaryotes and bacteria capable of modifying exogenous sterols can provide some insight into the types of molecules that might be present in these sterol-producing bacteria. This includes steryl glucosides, which have been identified in both eukaryotes[35] and sterol-modifying bacteria[39], and sterol esters, which to our knowledge have only been found in eukaryotes[16,42,43]. Expanding the extraction techniques used to analyze sterol lipids, identifying sterol conjugates present, and exploring enzymes responsible for downstream modifications would allow a better assessment of conjugated sterol diversity in the bacterial domain while providing a foundation for further exploration of sterol function.

Cholesterol production by bacteria also points to a complicated evolutionary history for sterol biosynthesis in the bacterial domain. Complex sterol biosynthesis is an ancient process; oxidosqualene cyclase (OSC), responsible for sterol cyclization, is thought to have evolved around the Great Oxidation Event[32] and modern-day eukaryotic sterol biosynthesis genes likely predate the last eukaryotic common ancestor[5]. Sterol biosynthesis in bacteria has been theorized to be a product of horizontal gene transfer from eukaryotes, supported by its rarity and uneven distribution across bacterial phyla[32]. However, recent phylogenetic and structural analyses of OSC and the C-14 demethylase (CYP51), suggest a bacterial origin for these proteins[25,26,44]. The putative *E. salina* cholesterol biosynthesis pathway we identified is largely homologous to the eukaryotic pathway, indicating a shared evolutionary history for much of cholesterol biosynthesis between this myxobacterium and eukaryotes, although our analysis does not provide insight into the directionality of acquisition. However, we did identify several cholesterol biosynthesis proteins involved in desaturation modification, and these proteins have not been considered in phylogenetic analyses (Supplementary Data 1). Further biochemical and phylogenetic analyses of these downstream proteins may better resolve the evolutionary relationships governing sterol production in these two domains.

Bacterial C-4 demethylation continues to provide a distinct example of independent evolution in sterol biosynthesis. We identified proteins responsible for complete C-4 demethylation in *E. salina* and *Calothrix* distinct from those in eukaryotes, aerobic methanotrophs, and each other. In eukaryotes, C-4 demethylation is an oxygen dependent reaction carried out by three proteins - a C-4 sterol methyl oxidase (ERG25/SMO), C-4 decarboxylase (ERG26/3β-HSD/D), and a 3-ketosteroid reductase (ERG27/3-SR)[45–47]. All three proteins are involved in an iterative process that sequentially removes both methyl

groups at the C-4 position although plants have been shown to utilize two distinct SMO proteins to remove each methyl group in a non-sequential manner[48]. We have previously shown that in aerobic methanotrophs C-4 demethylation results in the removal of one methyl group and is carried out by a Rieske-type oxygenase, SdmA, and an NADP-dependent reductase, SdmB[31]. Although both *E. salina* and *Calothrix* harbored homologs of SdmA and SdmB, it was unclear if these two proteins would be sufficient to remove both methyl groups at the C-4 position. We show that the two *Calothrix* homologs are indeed sufficient to remove both methyl groups at C-4 but *E. salina* requires a second reductase to fully demethylate. These enzymes are non-homologous to the canonical eukaryotic proteins and establish that cholesterol biosynthesis in these bacteria is not a simple case of horizontal gene transfer. Rather, it represents a case of convergent evolution in sterol biology that implies a critical role of C-4 sterol demethylation in sterol function in both domains of life. Indeed, C-4 demethylation has been shown to be required for proper function in eukaryotes as C-4 demethylase mutations are often lethal[45,49,50]. What remains unclear, is what functional role C-4 demethylation plays in bacterial cells and if it is required for proper sterol function as observed in eukaryotes. Further biochemical and structural characterization of the various bacterial C-4 demethylation pathways should provide comparative insights into the functional significance of this modification.

Finally, complex sterol biosynthesis in bacteria is also of interest for its potential industrial and biomedical applications. Production of cyclic triterpenoid lipids, including hopanoids in bacteria and sterols in fungi, have been suggested to increase the resiliency of these microbes to the different stresses encountered under industrial culturing conditions, including temperature and ethanol stress[51,52]. A better understanding of sterol biosynthesis genes and development of sterol expression systems provides further opportunities to better engineer microbes for industrial applications. Additionally, several myxobacteria, including other isolates of *E. salina*, produce steroid-derived secondary metabolites with demonstrated antimicrobial properties[53–55]. However, myxobacteria, particularly those from marine environments like *E. salina*, are often difficult to culture, genetically intractable, and produce secondary metabolites at low concentrations[56]. This has led to increased interest in understanding natural product biosynthesis pathways in these organisms and in development of heterologous systems to produce these compounds[57]. Exploration of steroid biosynthesis in myxobacteria may further reveal novel biosynthesis enzymes while providing a framework to better produce these compounds. Furthermore, the modular heterologous expression system we use here to explore C-4 demethylation presents an opportunity to overproduce biosynthetic intermediates in cholesterol biosynthesis, many of which are otherwise commercially unavailable or prohibitively expensive. These biosynthetic intermediates have proved useful for studying cholesterol biosynthesis, regulation, and function in eukaryotes and in some cases have clinical implications[58]. Continuing to characterize bacterial cholesterol biosynthesis using these heterologous systems will further build up tools to better study sterols in other organisms, while also providing insight into broader questions around sterol biology and evolution.

## Methods

### Bacterial culture
Strains used in this study are listed in *SI Appendix*, Table S4. *E. salina* DSM 15201 was cultured in 20 ml of Seawater Salts (SWS) medium, pH 7 in an Erlenmeyer flasks, supplemented with autoclaved and concentrated whole cell *E. coli*, at 30 °C with shaking at 225 rpm (Thermo Scientific, MaxQ8000), for 14 days[27]. To prepare the concentrated whole cell *E. coli*, *E. coli* DH10B was grown to OD$_{600}$ between 1.0 and 1.5 in 500 ml of LB media in an Erlenmeyer flask. This *E. coli* was pelleted, resuspended in 50 ml of SWS, and autoclaved. Over the course of

14 days, 5 ml of this concentrated *E. coli* suspension was fed to the *E. salina* culture as the media cleared[59]. *Calothrix* sp. NEIS-4105 was cultured in 20 ml of BG-11 liquid medium, pH 7[60] in an Erlenmeyer flask at 25 °C for 60 days with 10-h light, 14-h dark cycles. *E. coli* heterologous expression strains were cultured in 25 ml of TYGPN medium at 30 °C or 37 °C, shaking at 225 rpm and supplemented, if necessary, with gentamycin (15 μg/mL), kanamycin (30 μg/mL), carbenicillin (100 μg/mL), and/or chloramphenicol (20 μg/mL).

### Lipid extraction
Unbound sterols were extracted from lyophilized cell pellets through a modified Bligh-Dyer extraction[28]. Pellets were sonicated for 1 h in 10:5:4 (vol: vol: vol) methanol: dichloromethane (DCM): water. Lipids were then phase separated using two times the volume 1:1 (vol: vol) DCM: water. The organic phase was transferred and evaporated under N$_2$ gas yielding total lipid extracts (TLE). Where applicable, TLE was fractionated by polarity using Si chromatography through the following column scheme: 1.5 column volumes of hexanes, 2 column volumes of 8:2 (vol: vol) hexane: DCM, 2 column volumes of DCM, 2 column volumes 1:1 (vol: vol) DCM: ethyl acetate, 2 columns volumes ethyl acetate yielding an alkynes, non-polar, ketone, alcohol, and polar fraction, respectively[61].

Bound sterols were analyzed by hydrolyzing either lipid extracts, Si-gel chromatography fractions, or lyophilized cell pellets in 1 N HCl or KOH in methanol and heated at 75 °C for 3 h. Reactions were neutralized using KOH or HCl respectively and phases separated using twice the volume of 1:1 (vol: vol) DCM: water. The organic phase was transferred and evaporated under N$_2$ gas. Where applicable, butylated hydroxytoluene (BHT) was added to acid hydrolysis samples at a final concentration of 0.01% (wt: vol) of BHT: MeOH[62].

Cholesterol, desmosterol, zymosterol, and 25-hydroxycholesterol(25OCH) concentrations in samples were quantified using a standard curve ranging from 10ng-100ng. Sterol content was calculated using the standard curve and normalized to the dried cell weight of each sample. Additionally, the limit of detection for sterols on our mass spectrometer was determined to be between 1–5 ng by diluting a sterol standard mix.

### GC-MS analysis
All lipids were derivatized to trimethylsilyl ethers using 1:1 (vol: vol) Bis(trimethylsilyl)trifluoroacetamide: pyridine and heating at 70 °C for 1 h before analysis on an Agilent 7890B Series GC. Lipids were separated on a 60 m Agilent DB17HT column (60 m x 0.25 mm i.d. x 0.1 μm film thickness) with helium as the carrier gas at constant flow of 1.1 mL/min and programed as follows: 100 °C for 2 min; then 8 °C/min to 250 °C and held for 10 min; then 3 °C/min to 330 oC and held for 17 min. In all, 2 μL of each sample was injected in splitless mode at 250 °C. The GC was coupled to a 5977 A Series MSD with the ion source at 230 °C and operated at 70 eV in EI mode scanning from 50-850 Da in 0.5 s. Lipids were analyzed using Agilent MassHunter Qualitative Analysis (B.06.00) and identified based on retention time and spectra by comparison to previously confirmed laboratory standards, published spectra[31,63,64], and spectra deposited in the American Oil Chemists' Society (AOCS) Lipid Library (http://lipidlibrary.aocs.org/index.cfm) or the National Institute of Standards and Technology (NIST) databases.

### Molecular cloning techniques
Plasmids and oligonucleotides used in this study are described in Supplementary Table 5 and Supplementary Table 6. Oligonucleotides were purchased from Integrated DNA Technologies (Coralville, IA). Genomic DNA from *E. salina* was isolated using the GeneJET Genomic DNA Purification Kit (Thermo Scientific). Genomic DNA from *Calothrix* sp. NEIS-4105 was isolated using a phenol-chloroform extraction[65], where one volume of 25:24:1 phenol: chloroform: isoamyl alcohol (vol:

vol: vol) was added to biomass, centrifuged and the aqueous layer removed. This was repeated twice before precipitating genomic DNA using twice the volume of 70% ethanol. Plasmid DNA was isolated using the GeneJET Plasmid Miniprep Kit (Thermo Scientific). DNA fragments used during cloning were isolated using the GeneJET Gel Extraction Kit (Thermo Scientific). DNA was sequenced by ELIM Biopharm (Hayward, CA).

Plasmids were constructed by sequence and ligation independent cloning (SLIC)[66]. Briefly, complementary overhangs were created on gel-purified PCR product inserts and a restriction enzyme-linearized vector by incubation with T4 DNA polymerase (EMD Milipore) in the absence of nucleotides. This was followed by an annealing reaction and transformation without ligation. *E. coli* strains were transformed by electroporation using a MicroPulser Electroporator (BioRad) as recommended by the manufacturer.

### Heterologous expression
Sterol biosynthesis genes were overexpressed in *E. coli* from compatible plasmids with either an IPTG-inducible *lac* or arabinose-inducible *araBAD* promoters[33]. Heterologous expression strains were constructed as described in Supplementary Table 7. Genes of interest were expressed from the IPTG-inducible plasmid pSRKGm-*lac*UV5-rbs5 and/ or the arabinose-inducible plasmid pBAD1031K. *E. coli* strains were cultured at 37 °C, in 20 mL TYGPN medium supplemented with antibiotics (as necessary) until mid-exponential phase when expression was induced with 500 μM IPTG and 0.2% (wt: vol) arabinose for 30–40 h at 30 °C with shaking at 225 rpm, before harvest of cells.

### Bioinformatic analysis
To identify sterol biosynthesis genes in *E. salina* and *Calothrix*, we conducted a BLASTp search[67]. To be considered a putative sterol biosynthesis gene, we set an e-value cut-off of $1 \times e^{-30}$ and percent identity cutoff of 30. BLASTp search results are listed in Supplementary Table 2. To identify other bacteria which harbored the sterol biosynthesis genes we identified in *E. salina* and *Calothrix*, we first conducted a BLASTp search of bacteria in the genomic databases on the JGI IMG portal (https://img.jgi.doe.gov/) for OSC ($<1 \times e^{-50}$, 30%ID). Of this subset of bacteria, we conducted further BLASTp searches ($<1 \times e^{-50,}$ 30%ID) using identified proteins from *E. salina* and *Calothrix*[68,69]. A neighbor-joining tree of SdmBC homologs was generated by aligning protein sequences using MUSCLE in MEGA (11.0.10). The phylogenetic tree was then generated using the gamma model, four gamma rate categories and 500 bootstrap replicates.

### Genomic sequencing and mutation identification
Library preparation (Illumina DNA Prep Kit; San Diego, CA) and whole genome sequencing was performed by SeqCenter (SeqCenter; Pittsburgh, PA) and sequenced on an Illumina NextSeq 2000, producing 2x151bp reads. Bclconvert (v3.9.3) was used for demultiplexing, quality control, and trimming. To identify mutations in the serial passaged strain, variant calling was performed using breseq (v0.35.4)[70].

### Reporting summary
Further information on research design is available in the Nature Portfolio Reporting Summary linked to this article.

## Data availability
The data generated in this study are provided in the Supplementary Information and Source Data files. The JGI IMG (https://img.jgi.doe. gov/) gene IDs for sterol biosynthesis genes in *E. salina* and *Calothrix* are provided in Supplementary Table 2. The Source Data file includes the raw total ion chromatogram data used to generate main text and supplementary figures, as well as the raw mass spectrometry data used to generate supplementary spectra data. Source data are provided with this paper.

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

## Acknowledgements
We thank members of the Welander lab for helpful discussions. Funding for this study was provided by National Science Foundation Award 1919153 (to P.V.W).

## Author contributions
A.K.L, J.H.W., and P.V.W. designed research; A.K.L and J.H.W. performed research; A.K.L, J.H.W., and P.V.W analyzed data; and A.K.L, J.H.W., and P.V.W wrote the paper.

## Competing interests
The authors declare no competing interests.
