## [Peer Review File · Nature Communications]

De novo cholesterol biosynthesis in bacteriaReviewer #1 (Remarks to the Author):

The authors describe their many experiments trying to comprehend the complete sterol biosynthetic pathway in bacteria: They used the model strains *Enhygromyxa salina* and *Calothrix* sp. NIES-4105. The manuscript is well written with appropriate use of tables and figures. All experiments are very logical. Of course, it is of great interest to get a better insight into sterol biosynthesis in bacteria, especially because little is known about it. However the authors should describe in more detail the practical benefit of the study (outlook). From my side the article is suitable for publication in Nature Communications.

Enclosed are my remarks:

- LL64: Why exactly were these bacteria chosen? Why are they of interest or why are they particularly suitable?
- LL106: Very good that this was checked but the addition of butylhydroxytoluol (BHT) could also prevent oxidation (see Figure S5)
- LL173: Mitsche et al (eLife, 2015,4, e07999.) showed that the K-R and B-Pathway are not strictly separated, it depends more on the tissue and cell-type. As the authors observed both studied bacteria strains showed different favored pathways. This was also observed in molds and yeasts. Are there any differences between growth conditions (supplement) or under hypoxia?
- Is it possible to get a staining (fluorescence) of sterols and sterol conjugates in bacteria to see where the sterols are located? I think that would be a nice experiment, it's a bit strange that only conjugated sterols were observed in *Calothrix* because, I guess, the enzymes in the biosynthesis works only with the free sterols.
- How many milligrams of lyophilized sample were used? What are the LODs for the measurements? How many μg sterols per mg dry weight biomass were detected? An estimate or rough approximation is sufficient, as I am aware that the aim of the work was not to use a fully validated sterol analysis method, but merely to detect different sterols for the first time and to study the biosynthesis of bacteria.
- Have you carried out experiments with sterol biosynthesis inhibitors (e.g. statins, azoles/triazoles)? Have they an effect on the viability or morphology? How do they change the sterol pattern?
- Fig. 1: Why is the intensity of the detected sterols in the acid hydrolyzed sample lower than in the Blight Dyer sample? Have you got an idea what are the "dirt peaks" in b)? The peak shape of IV is strange? Are there two co-eluting sterols (see below: mass-deconvolution)? Why were 7 sterols detected in a) and only 5 sterols b)?
- Fig. 3: What is the composition of the sterol standard mix? Which concentration?
- Supporting Information: L10: Amount?
- Supporting Information: L10-32: Derivatization reagents, concentrations and procedure is missing!
- Supporting Information: Fig S1: Is the figure necessary? Good spectra are also available from Mueller et. al (Nature Protocols, 2019, 14, 2546). The use of the mass-deconvolution tool by Mass Hunter Software could result in better spectra (noise free spectra). I would prefer 4,4-dimethylcholesta-8,24-dienol and cholesta-7,24-dienol. Mass-deconvolution could also be useful for Fig. S8. Please provide detailed references for the identification of each sterol which are not included in the NIST spectral database (especially for the identification of sterols in Fig. S8).
- Supporting Information: Fig. S2: Is the use of the abbreviations ERG25-27 correct? I thought they are only used in yeasts.
- Supporting Information: Fig. S3: Use the same scale of the x-axis in all chromatograms (main text and Supporting info) Add a positive control or a standard mixture where "all" sterols were detected.
- Supporting Information: Fig. S4: Use "side chain" instead of "tail". The identification of oxysterols could work with the help of incremental rules or their retention behaviour. Basically, it can only be 25-hydroxycholest-7-enol or 25-hydroxycholest-8-enol. I find the mass spectrum of ** very strange. If M^+ is m/z 546 m/z 454 makes no sense because $M^+ - m/z$ 92 is not a typical fragmentation for sterols TMS ether. Normally a fragmentation of m/z -15 for methyl group, m/z -90 for $\text{HOSi}(\text{Me})_3$ or a m/z 105 is observed.

Reviewer #2 (Remarks to the Author):

The authors in the manuscript "De novo cholesterol biosynthesis in the bacterial domain" identified bacterial sterols in adapted *E. salina* to grow in liquid culture and *Calothrix* sp. NIES-4105 via GC-MS. The authors showed that acid hydrolysis of the lipid Bligh Dyer lipid extract enabled the detection of additional sterol molecules suggesting that the cleavage of ester- and ether-bonds after hydrolysis indicates the presence of sterol-bound molecules in both bacteria. By utilizing the mass spectra of known key sterol intermediates for cholesterol biosynthesis (By the Bloch pathway) and acid-hydrolysis of external standards cholesterol and desmosterol to indicate bacterial origin of these intermediates, the authors hypothesized about the preference of *E. salina* and *Calothrix* spp. for the Bloch and the dual Bloch/K-R sterol biosynthesis pathways, respectively. The authors performed a Blastp search for sterol biosynthesis proteins and found homologs of the eukaryotic cholesterol biosynthesis pathway in three *E. salina* strains supporting the argument of a Bloch pathway for sterol biosynthesis in these bacteria despite the unusual genomic localization of these genes for a bacterial biosynthetic pathway (array is not in operons). Blastp search in *Calothrix* spp genome showed different candidate enzymes compared to *E. salina* supporting a different mechanism for sterol biosynthesis among these 2 types of examined bacteria. In contrast to *E. salina* results, the authors found seven known sterol biosynthetic genes organized in 2 operons within *Calothrix* spp genome. Lastly, the authors characterized the enzymatic activity of SdmAB-C homologues from these bacteria in *E. coli* as experimental evidence of C4 demethylation due to these enzymes.

Overall, this manuscript presents some characterization data needed for the largely unknown and neglected area of bacterial sterol biosynthesis. The selected bacteria and the evidence presented here suggest substantial differences among bacterial sterol synthesis pathways of the examined bacteria. However, in occasions the evidence is insufficient to support the proposed pathways for sterol biosynthesis. The claim of that these bacteria can produce cholesterol de-novo (not even a bacterial cholesterol homologue or a cholesterol-like molecule) is concerning especially if the bacterial pathways are not fully elucidated in this study and bacterial pathways likely differ from those of eukaryotic origin.

Major comments

Line 83. The authors claimed for the first time that cholesterol was detected after B&D extraction from *E. salina* biomass. The detection method utilized here was a comparison of previously reported chromatograms and m/z spectra. An additional run with an external standard (cholesterol molecule) utilizing the settings for detection is needed for both Fig. 1 (*E. salina*) and Fig. 3 (*Calothrix* spp.). Alignment of both m/z spectra (sample) and experimentally run cholesterol (standard) will clearly demonstrate that the detection molecule is indeed cholesterol. If this molecule is a similar molecule to cholesterol this should be indicated as well, and the usage of the term cholesterol should be replaced throughout the text.

The chromatograms showed in Figure1 A,B showing the sterol species after conventional methanol or methanol-acid base treatments should be combined into either a single chromatogram with distinct color settings or in other array to clearly show the similar/different peaks after these 2 treatments. -Significant relative abundances showed up after 45 min in chromatogram after acid hydrolysis of *E. salina*, please indicate the nature of these unlabeled peaks.

Figure S5. Chromatograms in Fig S5 showing hydrolyzed cholesterol and desmosterol standards showed additional peaks compared to the hydrolysis without sterol standards. Labelling these molecules and showing the 25-hydroxycholesterol standard retention time and spectra (Fig S4) within in the same figure will strength the evidence for the presence of ether-bound 25-hydroxycholesterol in *E. salina*.

Figure 2a shows a potential Bloch pathway for cholesterol biosynthesis in *E. salina*. The authors showed in Figure 1 and supplemental the identification of all these molecules except for the potential product of C-14 demethylase: 4,4-dimethyl-cholest-8,14,24,-triene-3-ol, and the

subsequent C-4 partially demethylated molecule: 4-methyl(zymo)sterol. Please indicate this in the schematic.

While the homologues found through Blastp suggest the biosynthesis of a cholesterol homologue through a branch of the canonical MVA pathway, the activity of bacterial homologues of SMO/OSC are known to be non-specific for a single substrate, the same might be true for the rest of the bacterial homologue proteins. The possibility of unspecific enzymatic activity towards various substrates should be at least discussed and indicated in key schematics (e.g., Fig 2a, 4a)

Once the external standards spectra are shown and can be clearly compared to TLE+hydrolysed samples, the newly detection of cholesterol-like and 25-hydroxycholesterol molecules in these bacteria should be indicated in key the schematics (Fig 2a, 4a). Figures of molecules detected in this study should be bolded or to distinguish them from those hypothetical intermediates that were not detected here.

Fig4a. (sterol biosynthesis pathway in *Calothrix* spp.) shows the Bloch pathway also shown for *E. salina* (Figure 2a). However, from the eight molecules, only five were detected in the present study including cholesterol and 25-hydroxycholesterol which need to be confirmed experimentally via external standards.

Lines 172-175. The authors indicated detection of a typical intermediate from the Kandutsch-Russel (K-R) cholesterol pathway. Thus, the possibility that this pathway takes place in *Calothrix* spp should be included in the results and schematic from Figure 4a, especially after having measured differences for SDR-type reductases SdmB/C for C-4 demethylation compared to *E. salina*. Figure S2 should be utilized to indicate potential sterol biosynthetic pathways in *Calothrix* spp. since the evidence presented here is not definitive for neither Bloch nor KR pathways or even definitive to support that cholesterol, and not a close homologue, is being synthesized by these bacteria.

Related to last comment, Figure 4 shows seven genes known for sterol biosynthesis in *Calothrix* spp. organized in cluster however the proposed pathway by the authors in Fig 4a can only include two out of these seven identified genes. Most likely an alternative sterol biosynthetic pathway than the one shown in Figure 4a takes place for sterol biosynthesis in *Calothrix* spp.

The authors mention that serial passages resulted in ceased sterol production in *Calothrix* spp. The authors showed that the OSC homolog is active in *E. coli* and discussed that the found mutations (Line 185-186) were unlikely to cause the lack of sterol production. The authors should further discuss or provide additional supplementations in the media to explore some reasonable explanation about this issue. It's concerning that there is no explanation of why the same bacteria and growth condition produced different outcomes in terms of sterol production.

Minor comments

Line 58-59. The authors claim "analyses have yet to detect complex sterols in bacteria" however lipid analyses have shown the production of complex sterols from bacterial origin. For instance, in 1975, brassicasterol (which structure has the same backbone as cholesterol) was detected in *Calothrix* spp (name of article: "unsaponifiable matter of green and blue-green algal lipids as a factor of biochemical differentiation of their biomasses"). However, the refined characterizations, such as those shown in this manuscript, are lacking in literature. Please re-phrase and add existence literature about the reported sterols in these bacteria.

The authors evolved *E. salina* to grow in liquid culture when supplemented with lysates of *E. coli* to obtain sufficient biomass for sterols detection. Please indicate the amount of biomass needed for the lipid extraction and detection.

Figure 1, 4A: dehydrodesmosterol is misspelled.

Line 95-96. References supporting the argument for employing methanolic base vs methanolic acid hydrolysis to cleave phosphodiester/allyl ether bonds are needed. A schematic showing the

mechanism would be also helpful.

Discussion about potential ligands of detected bound-sterols, especially for the case of *Calothrix* spp, will be helpful.

The authors discussed about the potential existence of a novel cholesterol biosynthesis pathway in these bacteria. Some reports indicate that bacterial SHC (OSC homologue) catalyzes in one-step the cyclization of squalene. Thus, bio-informatic analyses between bacterial homologues of *E. salina* and *Calothrix* spp and bacterial SHC sequences will be helpful to discuss another key enzymatic step related to the convergent evolution of sterol biology (line 293).

Title needs to be more specific about the findings of this study. For example, the C4-demethylation characterized here is completely missing.

Discussion about the presence of the 2 operons with sterol biosynthetic genes in *Calothrix* spp (Fig 4b) and the absence of organized clusters for the sterol biosynthetic genes in *E. salina* (Fig S6) is needed to support a potential orchestrated activity of these enzymes at all within these bacterial species.

More information is needed about the adaptation of *E. salina* to growth in liquid medium. How many generations? Did the cells pass through stationary phase, or were they maintained in log phase? Was there a corresponding change in colony appearance, growth conditions, phenotype associated with the loss of cholesterol production?

An approximate lower limit of detection should be provided for the metabolite detection methods.

This reviewer disagrees with the authors' conclusion that "the loss of sterol production is unlikely to be caused by genetic mutation" (line 188). The authors need to provide some sort of example of how this loss of function could be due to something other than a genetic mutation. Just because the authors are unable to identify the specific mutation that caused this loss of function, it does not mean that a mutation is not the root cause.

Additional information is needed about the concentrated whole-cell *E. coli*. What strain of *E. coli* was used? How was it grown? Some sort of concentration needs to be provided to describe the relative abundance of the *E. coli* in the artificial seawater liquid media.

The authors need to either explicitly list the composition of the artificial seawater medium and BG-11 medium, or provide a supporting reference that describes these media types.

The authors need to specify the culture volume, initial pH, and vessel type.

Any concentrations provided as a % need clarification as to whether the concentration is vol%, mol%, wt%, wt/vol%, etc.

The pathways identified here may be relevant to the engineering of robust microbial cell factories. For example, Brenac et al 2019 "Distinct functional roles for hopanoid composition in the chemical tolerance of *Zymomonas mobilis*", Caspeta et al "Altered sterol composition renders yeast thermotolerant" (2014), Santoscoy "Production of cholesterol-like molecules impacts *Escherichia coli* robustness, production capacity and vesicle trafficking". The authors are encouraged to address this in their discussion.

Reviewer #3 (Remarks to the Author):

This paper provides the first strong evidence that certain bacteria synthesize cholesterol and related steroids. This is an important advance in our understanding of bacterial metabolism. The paper also provides insight to the synthetic pathway, which appears to be much like that of eukaryotes with some key differences. And, it provides some insight, but no strong conclusions, about the evolutionary history of the pathway. This work may contribute to the development of

biotechnology for steroid production.

The study is well executed, and provides conclusive evidence that *Enhygromyxa salina* synthesizes cholesterol and related steroids, and that its cells contain free steroids and steroid conjugates. However, evidence that *Calothrix* sp. produces similar steroids does not appear to be reproducible (further explained below). Heterologously expressed genes from both bacteria were shown to encode enzymes capable of some of the proposed reactions in cholesterol synthesis. The main methods used in this study were culturing of microorganisms, analysis and identification of steroids, and molecular genetics. This work was done to high standards with ample controls and other checks.

Following are specific comments and recommendations.

L180. The loss of the ability of *Calothrix* sp. to synthesize steroids during this study is very problematic. The genetic evidence and experiments strongly suggest that *Calothrix* sp. can do so, and the results from expression of *Calothrix* sp. genes adds important additional information. But, if the production of steroids by *Calothrix* sp. cannot be reproduced, it should not be reported. Did the authors go back to a stock culture and repeat the initial steroid (steroid conjugate) production experiment? If not, they must do so. If this cannot be reproduced, it should not be reported. But, it would still be possible to present the data for the *Calothrix* sp. gene expression experiments.

L96. Was the TLE hydrolyzed with methanolic acid further analyzed to try to identify the compounds that were conjugated with the steroids? Several new, large peaks appear in Fig. 1b after 45 min. Could any of these peaks represent lipids that were conjugated to sterols?

Fig S9. It is puzzling that peak I (lanosterol) is so much smaller in the lower panel than in the upper one. Is it possible that the construct in the lower panel metabolizes lanosterol to some product(s) that are not detected by the method used? Or, is there some reason that construct produces less lanosterol?

L94. It would be useful to compare in *E. salina* quantities of steroids per g cell biomass from direct extraction versus from extraction after acid hydrolysis. This would give an indication of the relative abundance of free steroids versus conjugates. Currently, it is unclear if both occur in substantial amounts in the cells.

L230. Given the inability to reproduce steroid conjugate synthesis in *Calothrix* sp., it is not appropriate to make comparisons about free versus conjugated steroids in *Calothrix* sp. versus *E. salina*.

L27. Perhaps add digestion as an important function (bile acids).

L60. The last paragraph of the Introduction recapitulates the main findings in the Abstract and elsewhere. It would be less redundant to focus here instead on the rationale and approach of the study.

L109. This paragraph would be clearer if a supplemental figure showed the cholesterol synthesis pathway, including SMO and OSC, and indicated the names and abbreviations used for the enzymes.

L133 states, "in the presence of lanosterol", which seems to imply exogenous lanosterol was added. It appears that the host strain overproduces lanosterol, so the text would be clearer without the words quoted here.

L138. A more accurate conclusion is that SdmAB in *E. salina* are not sufficient to fully demethylate the C-4 position "of lanosterol" (ie, they might do so to another steroid substrate).

Fig S6 does not add much to the manuscript. The statement that steroid synthesis genes are not in clusters in *E. salina* is sufficient without all the gene maps.

Figs 2 and 4 show the same biosynthetic pathway. To avoid this redundancy, consider a figure that combines the results for both bacterial strains. This would have the added benefit of facilitating

comparison of the two.

L154 should probably refer to Fig. S7.

Figs S1 and S8. There is redundancy between these figures. Probably they should be combined.

L123 is a confusing conclusion, because later it is shown that *Calothrix* sp. produces cholesterol, and Table S1 shows that it has homologs for most of the cholesterol synthesis genes in *E. salina*. Perhaps specify in L123 that the conclusion is "based on the stringent search criteria used".

L217. Conclusion not substantiated. Only one species was conclusively shown to produce cholesterol.

L219. Conclusion not substantiated; what evidence suggests a regulatory role for sterols in bacteria? Should clearly state that this is purely speculative.

L260. Change "existence of a potential novel cholesterol biosynthesis pathway" to "possibility of an alternate cholesterol biosynthesis pathway".

L267. "These missing homologs" is unclear. No missing homologs were previously discussed.

L278 states, "function through a mechanism separate from eukaryotes". Is this correct? Consider that nonhomologous genes may encode proteins with the same biochemical function/reaction mechanism. If reaction mechanisms are not known, it is probably more accurate to discuss how parts of the pathways are not homologous in the various taxonomic groups.

REVIEWER COMMENTS

Reviewer #1 (Remarks to the Author):

The authors describe their many experiments trying to comprehend the complete sterol biosynthetic pathway in bacteria: They used the model strains *Enhygromyxa salina* and *Calothrix* sp. NIES-4105. The manuscript is well written with appropriate use of tables and figures. All experiments are very logical. Of course, it is of great interest to get a better insight into sterol biosynthesis in bacteria, especially because little is known about it. However, the authors should describe in more detail the practical benefit of the study (outlook). From my side the article is suitable for publication in Nature Communications.

We thank the reviewer for their constructive comments.

Enclosed are my remarks:

- LL64: Why exactly were these bacteria chosen? Why are they of interest or why are they particularly suitable?

Both our own preliminary analysis and several recent phylogenetic studies have identified a genetic potential for complex sterol biosynthesis beyond what has been previously observed in *E. salina* and *Calothrix*. In the case of *E. salina*, we had previously identified zymosterol in biomass scraped off plates. However, these growth conditions resulted in low biomass recovery, limiting analyses. Given we had identified nearly all the homologs required for cholesterol biosynthesis in the *E. salina* genome, the discrepancy between the sterols we had previously observed and the apparent genomic capacity motivated us to re-analyze sterol in this bacterium.

To better capture this in the introduction we have edited the final paragraph of the introduction to focus on the motivation and rationale behind our study.

- LL106: Very good that this was checked but the addition of butylhydroxytoluol (BHT) could also prevent oxidation (see Figure S5)

We have added this additional control to the acid hydrolysis control supplementary figure (Fig S6).

- LL173: Mitsche et al (eLife, 2015,4, e07999.) showed that the K-R and B-Pathway are not strictly separated, it depends more on the tissue and cell-type. As the authors observed both studied bacteria strains showed different favored pathways. This was also observed in molds and yeasts. Are there any differences between growth conditions (supplement) or under hypoxia?

We have only tested how growth conditions impact sterol production in a limited capacity (tested conditions include temperature, pH, ethanol stress, and for *E. salina* starvation to induce sporulation/fruiting body formation) and note that *E. salina* (as well as other Myxobacteria) in particular are very challenging to cultivate, let alone subject to additional physiological stress. We are still in the process of assessing the impacts on total sterol production under these different conditions but our initial non-quantitative assessments do not show any dramatic changes in sterol production or pathway usage under these conditions. This is currently the focus of a follow-up study.

- Is it possible to get a staining (fluorescence) of sterols and sterol conjugates in bacteria to see where the sterols are located? I think that would be a nice experiment, it's a bit strange that only conjugated sterols were observed in *Calothrix* because, I guess, the enzymes in the biosynthesis works only with the free sterols.

Sterol localization in bacteria is an interesting question and particularly in myxobacteria like *E. salina* and heterocyst-forming cyanobacteria like *Calothrix* which are both capable of differentiation into different cell types. Fluorescence staining of sterols in *E. salina* is particularly appealing as most commercially available stains target more modified sterols like cholesterol. These are experiments that we are planning to pursue in the future but we think they are beyond the scope of this study.

- How many milligrams of lyophilized sample were used? What are the LODs for the measurements? How many μg sterols per mg dry weight biomass were detected? An estimate or rough approximation is sufficient, as I am aware that the aim of the work was not to use a fully validated sterol analysis method, but merely to detect different sterols for the first time and to study the biosynthesis of bacteria.

We added a supplementary table (Table S1) detailing the dry weight of biomass used in extractions as well as the concentration of cholesterol, desmosterol, and zymosterol detected. Because the response factor varies between sterols, we did not include rough estimates for sterols that we did not have standards for. We also added information about the limits of detection to the supplementary methods.

- Have you carried out experiments with sterol biosynthesis inhibitors (e.g. statins, aoles/triazoles)? Have they an effect on the viability or morphology? How do they change the sterol pattern?

We have cultured *E. salina* in the presence of terbinafine, an allylamine that blocks all sterol biosynthesis by inhibiting squalene monooxygenase. Terbinafine inhibits growth of *E. salina*, suggesting sterols may be essential to this bacterium. However, we have been unable to recover *E. salina* cultures grown with terbinafine by adding exogenous sterols (exogenously added sterols include lanosterol, desmosterol, cholesterol, and 25-hydroxycholesterol). These

results could be due to number of factors including insufficient sterol delivery/uptake (*E. salina* grows in aggregated clumps in liquid culture), use of wrong supplemental sterols (our data suggests *E. salina* further modifies its sterols and these modified sterol compounds might be what's needed to recover cultures), or off-target effects of terbinafine. These experiments are part of on-going studies in our group on the physiology of sterols in myxobacteria.

- Fig. 1: Why is the intensity of the detected sterols in the acid hydrolyzed sample lower than in the Bligh Dyer sample? Have you got an idea what are the “dirt peaks” in b)? The peak shape of IV is strange? Are there two co-eluting sterols (see below: mass-deconvolution)? Why were 7 sterols detected in a) and only 5 sterols b)?

We observe both loss of sterols through the lipid extraction protocol and degradation of sterols when acid hydrolyzed. The sterols in acid hydrolysis chromatogram presented in figure 1 were first extracted using a Bligh Dyer, then purified using Si-column chromatography before treating with methanolic acid. We think we lose sterols to each step in the extraction process. We think this is also why we only detect those sterols that are at higher abundance in the acid hydrolysis samples. We haven't been able to identify the peaks in b) that come off after the 45-minute mark, however we suspect they are likely modified sterols based on the spectra and we've now included these spectra as a part of supplemental figure S5. Given a mass ion of 488 (which could be achieved by adding an ether to either the C-3 or C-25 hydroxyl) and a prominent 73 peak (which is also the only real diagnostic peak of cholesterol diesters).

Peak IV is co-eluting with another sterol. Using the deconvolution tool in the GC-MS data analysis software (Mass Hunter) we were able to get a cleaner spectra for this compound that better reflects the spectra previously reported in the literature.

- Fig. 3: What is the composition of the sterol standard mix? Which concentration?

The sterol standard mix includes cholesterol, desmosterol, zymosterol, lanosterol, and 25-hydroxycholesterol. 40ng of each sterol was injected into the GC-MS. We added this as clarifying text to the legend of figure 3.

- Supporting Information: L10: Amount?

We add a supplementary table (Table S1) detailing the amount of sterols extracted and the dry weight of bacterial biomass the sterols were extracted from.

- Supporting Information: L10-32: Derivatization reagents, concentrations and procedure is missing!

We added derivatization procedure to supplementary methods, subsection lipid extractions.

- Supporting Information: Fig S1: Is the figure necessary? Good spectra are also available from Mueller et. al (Nature Protocols, 2019, 14, 2546). The use of the mass-deconvolution tool by Mass Hunter Software could result in better spectra (noise free spectra). I would prefer 4,4-dimethylcholesta-8,24-dienol and cholesta-7,24-dienol. Mass-deconvolution could also be useful for Fig. S8. Please provide detailed references for the identification of each sterol which are not included in the NIST spectral database (especially for the identification of sterols in Fig. S8).

We think that, while reported elsewhere in the literature, including the spectra for the sterols we identified provides important supporting evidence for our identification and increases transparency around sterol analysis in the field broadly. We have re-extracted spectra after using the mass-deconvolution tool as suggested. We also have included references for spectra from the literature in the supplementary methods.

- Supporting Information: Fig. S2: Is the use of the abbreviations ERG25-27 correct? I thought they are only used in yeasts.

The ERG designation does refer specifically to the yeast homologs for these steps, although they are homologous to the proteins found in animals and plants. To further clarify this, we added the animal homolog designation to supplementary figure S12. We have changed the supplement cholesterol biosynthesis pathway to focus on *E. salina* and *Calothrix* specifically and it no longer has eukaryotic protein labels for C-4 demethylation.

- Supporting Information: Fig. S3: Use the same scale of the x-axis in all chromatograms (main text and Supporting info) Add a positive control or a standard mixture where “all” sterols were detected.

We adjusted x-axis on the supplementary chromatograms, so they matched the chromatograms in the main text. We also added an additional supplementary figure showing our sterol standard mix run alongside sterols extracted from cell biomass (figure S2).

- Supporting Information: Fig. S4: Use “side chain” instead of “tail”. The identification of oxysterols could work with the help of incremental rules or their retention behaviour. Basically, it can only be 25-hydroxycholest-7-enol or 25-hydroxycholest-8-enol. I find the mass spectrum of ** very strange. If M+ is m/z 546 m/z 454 makes no sense because M+ -m/z 92 is not a typical fragmentation for sterols TMS ether. Normally a fragmentation of m/z -15 for methyl group, m/z -90 for HOSi(Me)₃ or a m/z 105 is observed.

It does make a lot of sense for the probable oxysterol to be either 25-hydroxycholesta-7-enol or 25-hydroxycholesta-8-enol given the mass of the compound and the sterol intermediates we

observe in *E. salina*. However, the spectra for these hydroxysterols do not have very many diagnostic peaks and without standards, we do not feel confident in identifying this compound. We reassessed the spectra of the second potential oxysterol. M+ is 544 which should be inline with the fragmentation patterns for sterols.

Reviewer #2 (Remarks to the Author):

The authors in the manuscript “De novo cholesterol biosynthesis in the bacterial domain” identified bacterial sterols in adapted *E. salina* to grow in liquid culture and *Calothrix* sp. NIES-4105 via GC-MS. The authors showed that acid hydrolysis of the lipid Bligh Dyer lipid extract enabled the detection of additional sterol molecules suggesting that the cleavage of ester- and ether-bonds after hydrolysis indicates the presence of sterol-bound molecules in both bacteria. By utilizing the mass spectra of known key sterol intermediates for cholesterol biosynthesis (By the Bloch pathway) and acid-hydrolysis of external standards cholesterol and desmosterol to indicate bacterial origin of these intermediates, the authors hypothesized about the preference of *E. salina* and *Calothrix* spp. for the Bloch and the dual Bloch/K-R sterol biosynthesis pathways, respectively. The authors performed a Blastp search for sterol biosynthesis proteins and found homologs of the eukaryotic cholesterol biosynthesis pathway in three *E. salina* strains supporting the argument of a Bloch pathway for sterol biosynthesis in these bacteria despite the unusual genomic localization of these genes for a bacterial biosynthetic pathway (array is not in operons). Blastp search in *Calothrix* spp genome showed different candidate enzymes compared to *E. salina* supporting a different mechanism for sterol biosynthesis among these 2 types of examined bacteria. In contrast to *E. salina* results, the authors found seven known sterol biosynthetic genes organized in 2 operons within *Calothrix* spp genome. Lastly, the authors characterized the enzymatic activity of SdmAB-C homologues from these bacteria in *E. coli* as experimental evidence of C4 demethylation due to these enzymes.

Overall, this manuscript presents some characterization data needed for the largely unknown and neglected area of bacterial sterol biosynthesis. The selected bacteria and the evidence presented here suggest substantial differences among bacterial sterol synthesis pathways of the examined bacteria. However, in occasions the evidence is insufficient to support the proposed pathways for sterol biosynthesis. The claim of that these bacteria can produce cholesterol de-novo (not even a bacterial cholesterol homologue or a cholesterol-like molecule) is concerning especially if the bacterial pathways are not fully elucidated in this study and bacterial pathways likely differ from those of eukaryotic origin.

We thank the reviewer for their constructive comments.

Major comments

Line 83. The authors claimed for the first time that cholesterol was detected after B&D extraction from *E. salina* biomass. The detection method utilized here was a comparison of previously reported chromatograms and m/z spectra. An additional run with an external standard (cholesterol molecule) utilizing the settings for detection is needed for both Fig. 1 (*E. salina*) and Fig. 3 (*Calothrix* spp.). Alignment of both m/z spectra (sample) and experimentally run cholesterol (standard) will clearly demonstrate that the detection molecule is indeed cholesterol. If this molecule is a similar molecule to cholesterol this should be indicated as well, and the usage of the term cholesterol should be replaced throughout the text.

To further support our identification of cholesterol and several cholesterol intermediates, we included a supplementary figure showing the chromatograms of sterols extracted from *E. salina* and *Calothrix* alongside a chromatogram of a sterol standard mix (cholesterol, desmosterol, zymosterol, lanosterol, and 25-hydroxycholesterol at 80ng). Spectra corresponding to both the cholesterol standard and the cholesterol observed in *E. salina* and *Calothrix* are included.

The chromatograms showed in Figure 1 A,B showing the sterol species after conventional methanol or methanol-acid base treatments should be combined into either a single chromatogram with distinct color settings or in other array to clearly show the similar/different peaks after these 2 treatments. -Significant relative abundances showed up after 45 min in chromatogram after acid hydrolysis of *E. salina*, please indicate the nature of these unlabeled peaks.

We tried combining or arraying the chromatograms in Figure 1 but feel because of the complexity of the samples, combining the chromatograms made the data harder to read. Supplementary figure two now includes chromats from figure 1, supplementary figure 14, and a sterol standard mix aligned to allow for easier comparison of sterols present in each condition.

To address the compounds that come off after the 45 min mark in acid hydrolyzed samples, we included an additional supplementary figure showing spectra of these compounds. While we are unable to identify these compounds, we suspect they are likely some sort of steroid. With a mass ion of 488 (which could be achieved by adding an ether to the C-3 or C25 position) and a prominent 73 peak (which is also found in sterol diethers) these compounds may represent an intact sterol ether. However, the spectrum does not really have diagnostic peaks and we are unable to find a sterol ether standard to compare to. We included the spectra of these compounds as putative sterols in supplementary figure S5.

Figure S5. Chromatograms in Fig S5 showing hydrolyzed cholesterol and desmosterol standards showed additional peaks compared to the hydrolysis without sterol standards. Labelling these molecules and showing the 25-hydroxycholesterol standard retention time and spectra (Fig S4) within in the same figure will strength the evidence for the presence of ether-bound 25-hydroxycholesterol in *E. salina*.

We have added the chromatogram showing a sterol standard mixture to this figure and labeled the non-sterol compounds. As an additional control against auto-oxidation we hydrolyzed *E. salina* biomass with butylated hydroxytoluene.

Figure 2a shows a potential Bloch pathway for cholesterol biosynthesis in *E. salina*. The authors showed in Figure 1 and supplemental the identification of all these molecules except for the potential product of C-14 demethylase: 4,4-dimethyl-cholest-8,14,24,-triene-3-ol, and the subsequent C-4 partially demethylated molecule: 4-methyl(zymo)sterol. Please indicate this in the schematic.

With how we have restructured the manuscript, figure 2a is now a standalone figure. The names of sterols identified in *E. salina* extracts have been bolded to indicate their observed presence. Additionally, we combined the two pathways found in figure 2a and 4a into a single pathway figure in the supplement (Figure S3) to make comparing putative biosynthesis pathways in these two bacteria easier and, we bolded the names of identified structures.

While the homologues found through Blastp suggest the biosynthesis of a cholesterol homologue through a branch of the canonical MVA pathway, the activity of bacterial homologues of SMO/OSC are known to be non-specific for a single substrate, the same might be true for the rest of the bacterial homologue proteins. The possibility of unspecific enzymatic activity towards various substrates should be at least discussed and indicated in key schematics (e.g., Fig 2a, 4a)

This is an interesting point in that several of the cholesterol biosynthesis proteins in eukaryotes have also been shown to work on multiple substrates *in vitro*. Additionally, the C-24 reductase has several functions outside of sterol biosynthesis including apoptosis and oxygen stress. It's very well possible the bacterial homologs we identified can also carry out reactions on multiple substrates. Further biochemical characterization of these bacterial homologs is of course needed to address these questions. We have modified the text in the results section to emphasize this point.

Once the external standards spectra are shown and can be clearly compared to TLE+hydrolysed samples, the newly detection of cholesterol-like and 25-hydroxycholesterol molecules in these bacteria should be indicated in key the schematics (Fig 2a, 4a). Figures of molecules detected in this study should be bolded or to distinguish them from those hypothetical intermediates that were not detected here.

We bolded the names of pathway intermediates detected in bacterial biomass for both *E. salina* and *Calothrix* in pathway figures where applicable and added clarifying text to the figure legend.

Fig4a. (sterol biosynthesis pathway in *Calothrix* spp.) shows the Bloch pathway also shown for

E. salina (Figure 2a). However, from the eight molecules, only five were detected in the present study including cholesterol and 25-hydroxycholesterol which need to be confirmed experimentally via external standards.

Given the difficulty reviewers have had with the loss of sterol detection in our *Calothrix* strains, we have restructured the manuscript to de-emphasize *Calothrix* sterol analysis (see comments below). This pathway figure was therefore moved to the supplement and this figure now includes both the Bloch and K-R pathways.

Lines 172-175. The authors indicated detection of a typical intermediate from the Kandutsch-Russel (K-R) cholesterol pathway. Thus, the possibility that this pathway takes place in *Calothrix* spp should be included in the results and schematic from Figure 4a, especially after having measured differences for SDR-type reductases SdmB/C for C-4 demethylation compared to *E. salina*. Figure S2 should be utilized to indicate potential sterol biosynthetic pathways in *Calothrix* spp. since the evidence presented here is not definitive for neither Bloch nor KR pathways or even definitive to support that cholesterol, and not a close homologue, is being synthesized by these bacteria.

This pathway figure has now been moved to the supplement (figure S2) and includes both the Bloch and K-R pathway. The names of the sterols identified by our analysis are bolded to further illustrate the putative nature of our proposed pathway.

Related to last comment, Figure 4 shows seven genes known for sterol biosynthesis in *Calothrix* spp. organized in cluster however the proposed pathway by the authors in Fig 4a can only include two out of these seven identified genes. Most likely an alternative sterol biosynthetic pathway than the one shown in Figure 4a takes place for sterol biosynthesis in *Calothrix* spp. Given our restructuring of the manuscript, Figure 4 no longer contains a proposed pathway for *Calothrix*. The proposed *Calothrix* pathways we show in the supplement are modeled after the cholesterol biosynthesis pathways found in eukaryotes and are meant to illustrate that *Calothrix* is likely synthesizing cholesterol through a possible alternative pathway. The seven homologs found in the *Calothrix* sterol biosynthetic gene cluster are all represented in these pathways so we are unclear why the reviewer states that the proposed pathway can only include 2 of these 7 identified genes. We suspect other genes localized in this cluster are likely responsible for carrying out the remaining steps in cholesterol biosynthesis, however without further biochemical characterization of these proteins, this remains unclear.

The authors mention that serial passages resulted in ceased sterol production in *Calothrix* spp. The authors showed that the OSC homolog is active in *E. coli* and discussed that the found mutations (Line 185-186) were unlikely to cause the lack of sterol production. The authors should further discuss or provide additional supplementations in the media to explore some reasonable explanation about this issue. It's concerning that there is no explanation of why the

same bacteria and growth condition produced different outcomes in terms of sterol production.

To better address the inconclusive nature of our lipid analysis in *Calothrix*, we have reorganized the sections of the paper. Our heterologous expression experiments data for *Calothrix* now falls in a single section with the *E. salina* heterologous expression data. We now include a short paragraph explaining our lipid extraction results but the corresponding chromatogram has been moved to the supplement and we end the paragraph by explicitly stating our sterol analysis in *Calothrix* is inconclusive. We still think there is merit in showing that heterologous expression showed the *Calothrix* genes are capable of generating (*osc*) and modifying sterols (*SdmAB*), that we did see sterol production, and that sterol production was lost. We have also removed sections from the discussion comparing the sterols present in *E. salina* and *Calothrix* and added context around previous sterol analysis in cyanobacteria.

Minor comments

Line 58-59. The authors claim “analyses have yet to detect complex sterols in bacteria” however lipid analyses have shown the production of complex sterols from bacterial origin. For instance, in 1975, brassicasterol (which structure has the same backbone as cholesterol) was detected in *Calothrix* spp (name of article: “unsaponifiable matter of green and blue-green algal lipids as a factor of biochemical differentiation of their biomasses”). However, the refined characterizations, such as those shown in this manuscript, are lacking in literature. Please rephrase and add existence literature about the reported sterols in these bacteria.

There were a handful of papers in the 70s that found complex sterols in cyanobacteria. However, at the time there was no supporting genomic evidence for sterol biosynthesis and these microbial cultures were not necessarily axenic. In Paoletti et al, 1975, their *Calothrix* sp. was cultured in a vessel outside and they explicitly state there was contamination in their cultures. Sterol production in cyanobacteria was revisited by Summons et al, 2006 and they demonstrated that, at least in some cases, the observed sterols were the product of fungal contamination, in agreement with a lack of sterol biosynthesis genes present in these cyanobacterial genomes. Since this study, sterol production in cyanobacteria has been largely discounted and never reexamined. While our sterol analysis in *Calothrix* was ultimately inconclusive, it does highlight the cyanobacteria as a group where sterol biosynthesis should be further investigated. We have included discussion about this controversy in the discussion section.

Outside of these early studies in cyanobacteria, to our knowledge, the only other confirmed reports of complex sterol biosynthesis in the bacteria domain are in the myxobacteria, which have been shown to produce zymosterol. This is still a few steps shy of the sterols typically observed in eukaryotes.

The authors evolved *E. salina* to grow in liquid culture when supplemented with lysates of *E.*

coli to obtain sufficient biomass for sterols detection. Please indicate the amount of biomass needed for the lipid extraction and detection.

We added the dry weight for the biomass extracted in supplementary table S1.

Figure 1, 4A: dehydrodesmosterol is misspelled.

We fixed the misspellings.

Line 95-96. References supporting the argument for employing methanolic base vs methanolic acid hydrolysis to cleave phosphodiester/allyl ether bonds are needed. A schematic showing the mechanism would be also helpful.

We have added references for hydrolysis of sterol ethers using acid.

Discussion about potential ligands of detected bound-sterols, especially for the case of *Calothrix* spp, will be helpful.

Based on the sterol conjugates observed in both eukaryotes and bacteria that modify exogenously acquired sterols, we would hypothesize that the sterol conjugates present in *Calothrix* and *E. salina* might be steryl glucoside or sterol ester-like compounds. We added text about the potential conjugates that may be present to the discussion.

The authors discussed about the potential existence of a novel cholesterol biosynthesis pathway in these bacteria. Some reports indicate that bacterial SHC I(OSC homologue) catalyzes in one-step the cyclization of squalene. Thus, bio-informatic analyses between bacterial homologues of *E. salina* and *Calothrix* spp and bacterial SHC sequences will be helpful to discuss another key enzymatic step related to the convergent evolution of sterol biology (line 293). Cyclase evolution is a very active field of research. However, the phylogeny of *osc* from both *E. salina* and *Calothrix* has already been considered by a few recent papers (Hoshino Y, Gaucher EA. Proc Natl Acad Sci U S A. 2021,118(25):e2101276118; Santana-Molina C, Rivas-Marin E, Rojas AM, Devos DP. Mol Biol Evol. 2020, 37(7):1925-1941; Gold DA, Caron A, Fournier GP, Summons RE. Nature. 2017, 543(7645):420-423) and we feel this type of in-depth phylogenetic analyses is beyond the scope of this study.

Title needs to be more specific about the findings of this study. For example, the C4-demethylation characterized here is completely missing.

We think the main finding of our work is the biosynthesis of cholesterol by *Enhygromyxa* and our work characterizing C-4 demethylation is inherent to understanding biosynthesis. We feel the title as is reflects these main finding.

Discussion about the presence of the 2 operons with sterol biosynthetic genes in *Calothrix* spp

(Fig 4b) and the absence of organized clusters for the sterol biosynthetic genes in *E. salina* (Fig S6) is needed to support a potential orchestrated activity of these enzymes at all within these bacterial species.

We are a little unclear as to what the reviewer is asking here. We agree that it is striking that *Calothrix* seems to contain sterol synthesis genes in a single gene cluster while *E. salina* has these genes spread throughout the genome. While the presence of a single gene cluster in *Calothrix* might suggest these genes are similarly regulated and transcribed together and the lack of organized gene clusters may point to a more complex regulatory network in *E. salina*, we feel that experiments detailing the transcriptional regulation mechanisms underlying bacterial sterol biosynthesis are beyond the scope of this current study.

More information is needed about the adaptation of *E. salina* to growth in liquid medium. How many generations? Did the cells pass through stationary phase, or were they maintained in log phase? Was there a corresponding change in colony appearance, growth conditions, phenotype associated with the loss of cholesterol production?

Liquid cultures of *E. salina* were inoculated from swarms grown on agar plates. These liquid cultures grew in clumps as has been reported with similar myxobacteria. No serial passaging was required to achieve this. We removed the language that suggests actual adaptation was required for growth in liquid culture and added references for the growth techniques used to culture this myxobacterium.

An approximate lower limit of detection should be provided for the metabolite detection methods.

We included the limits of detection for sterols in the supplementary methods.

This reviewer disagrees with the authors' conclusion that "the loss of sterol production is unlikely to be caused by genetic mutation" (line 188). The authors need to provide some sort of example of how this loss of function could be due to something other than a genetic mutation. Just because the authors are unable to identify the specific mutation that caused this loss of function, it does not mean that a mutation is not the root cause.

We have reorganized and rewritten the paper to de-emphasize our analysis of sterols in *Calothrix* and now only state that there are no mutations in the sterol biosynthesis gene cluster found in *Calothrix*.

Additional information is needed about the concentrated whole-cell *E. coli*. What strain of *E. coli* was used? How was it grown? Some sort of concentration needs to be provided to describe the relative abundance of the *E. coli* in the artificial seawater liquid media.

The *E. coli* strain used as a feed stock to culture *E. salina* was DH10B. We grew 500ml of *E. coli* to an OD of 1.0-1.5 and pelleted and resuspended the *E. coli* in 50ml of the SWS medium used to culture *E. salina*. To decrease opportunities for contamination and ensure the *E. coli* stock did not continue to grow and change the stock conditions, we autoclaved the concentrated *E. coli*. 5ml of this concentrated suspension was added to each *E. salina* culture over 14 days of growth, as the *E. salina* culture cleared the *E. coli* from the media. Similar methods have been used to culture myxobacteria by other groups. We have added this additional information to the bacterial culture section in the methods.

The authors need to either explicitly list the composition of the artificial seawater medium and BG-11 medium, or provide a supporting reference that describes these media types.

We added references which include the composition of the sea water media used to culture *E. salina* and BG-11 used to culture freshwater cyanobacteria to the bacterial culture section of the methods.

The authors need to specify the culture volume, initial pH, and vessel type.

We added this additional information to the bacterial culture section of the methods.

Any concentrations provided as a % need clarification as to whether the concentration is vol%, mol%, wt%, wt/vol%, etc.

We added clarifying text where needed.

The pathways identified here may be relevant to the engineering of robust microbial cell factories. For example, Brenac et al 2019 “Distinct functional roles for hopanoid composition in the chemical tolerance of *Zymomonas mobilis*”, Caspeta et al “Altered sterol composition renders yeast thermotolerant” (2014), Santoscoy “Production of cholesterol-like molecules impacts *Escherichia coli* robustness, production capacity and vesicle trafficking”. The authors are encouraged to address this in their discussion.

That’s a great point! Additionally, our work has some potential biomedical applications as several myxobacteria, including other isolates of *E. salina*, are known to produce novel antibiotics that are derived from steroids. However, these myxobacteria are difficult to culture, genetically intractable, and typically produce these compounds at low concentrations. Because of this, there is some interest in identifying the biosynthetic pathways driving production of these molecules and recapitulating them in heterologous systems. We have added an additional paragraph to the discussion section to address these points.

Reviewer #3 (Remarks to the Author):

This paper provides the first strong evidence that certain bacteria synthesize cholesterol and related steroids. This is an important advance in our understanding of bacterial metabolism. The paper also provides insight to the synthetic pathway, which appears to be much like that of eukaryotes with some key differences. And, it provides some insight, but no strong conclusions, about the evolutionary history of the pathway. This work may contribute to the development of biotechnology for steroid production.

The study is well executed, and provides conclusive evidence that *Enhygromyxa salina* synthesizes cholesterol and related steroids, and that its cells contain free steroids and steroid conjugates. However, evidence that *Calothrix* sp. produces similar steroids does not appear to be reproducible (further explained below). Heterologously expressed genes from both bacteria were shown to encode enzymes capable of some of the proposed reactions in cholesterol synthesis. The main methods used in this study were culturing of microorganisms, analysis and identification of steroids, and molecular genetics. This work was done to high standards with ample controls and other checks.

Following are specific comments and recommendations.

L180. The loss of the ability of *Calothrix* sp. to synthesize steroids during this study is very problematic. The genetic evidence and experiments strongly suggest that *Calothrix* sp. can do so, and the results from expression of *Calothrix* sp. genes adds important additional information. But, if the production of steroids by *Calothrix* sp. cannot be reproduced, it should not be reported. Did the authors go back to a stock culture and repeat the initial steroid (steroid conjugate) production experiment? If not, they must do so. If this cannot be reproduced, it should not be reported. But, it would still be possible to present the data for the *Calothrix* sp. gene expression experiments.

We thank the reviewer for these thoughtful suggestions. We agree that the loss of sterol synthesis by *Calothrix* is problematic, and we struggled with how best to present this data. Genomic evidence and our initial lipid analyses provide strong evidence that sterol synthesis is carried out by this organism. Not reporting this to us seemed a disservice – others may see these genes in the genome and attempt to detect sterols and may detect them or not. If we did not have genomic evidence and the heterologous expression data, we would doubt our lipid results more but taken together, these data, in our opinion, point to a loss of sterol synthesis in this organism after serial passage.

We would have liked to reorder the original strain from the strain databank, however, they no longer have an active culture of this particular *Calothrix* sp. Given the biosynthetic gene cluster is still intact, we have tried varying the conditions of growth to try and reinduce cholesterol production, but we have yet to find conditions that result in any sterol production.

Sterol production in cyanobacteria has a history of being controversial and inconclusive. Early analyses in the 1970s before there was supporting genomic evidence suggested these bacteria

could produce highly modified sterols, specifically those with C-24 methylations typical of plants and fungi. Later work by Summons et al (2006) found that in several cases the sterol observed in these cyanobacteria was due to fungal contamination and this was supported by a lack of sterol biosynthesis genes in the genomes of these particular cyanobacteria. Since this 2006 study, sterol production has not been revisited.

We agree that our analysis of sterol in *Calothrix* is inconclusive. However, given that we did initially see sterols in this bacterium, partially supported by the genes required to produce fully demethylated sterols, we feel it would be disingenuous to exclude this data while also missing an opportunity to highlight the cyanobacteria as group where sterol biosynthesis should be further investigated.

To better address the inconclusive nature of our lipid analysis in *Calothrix*, we have reorganized the sections of the paper. Our heterologous expression experiments data for *Calothrix* now falls in a single section with the *E. salina* heterologous expression data. We now include a short paragraph explaining our lipid extraction results but the corresponding chromatogram has been moved to the supplement and we end the paragraph by explicitly stating our sterol analysis in *Calothrix* is inconclusive. We have also removed sections from the discussion comparing the sterols present in *E. salina* and *Calothrix* and added context around previous sterol analysis in cyanobacteria.

L96. Was the TLE hydrolyzed with methanolic acid further analyzed to try to identify the compounds that were conjugated with the steroids? Several new, large peaks appear in Fig. 1b after 45 min. Could any of these peaks represent lipids that were conjugated to sterols?

We have yet to do any further analysis to identify either intact sterol conjugates or the degraded conjugate products. The spectra of the peaks present after the 45 minute mark would suggest they are sterols. We would speculate that with a mass ion of 488 (which could be achieved by adding an ether to the C-3 or C25 position) and a prominent 73 peak (which is the characteristic peak in cholesterol diether) these compounds may represent an intact sterol ether. We have included the spectra of these compounds in supplementary fig S5.

Fig S9. It is puzzling that peak I (lanosterol) is so much smaller in the lower panel than in the upper one. Is it possible that the construct in the lower panel metabolizes lanosterol to some product(s) that are not detected by the method used? Or, is there some reason that construct produces less lanosterol?

We are unsure why there is less lanosterol in this experiment. This is expression of the aerobic methanotroph SdmA homolog with the *E. salina* SdmB homolog that showed no demethylation. It is possible that expression of these two proteins together causes some conversion of lanosterol to a product that we did not detect. It could also be that over-expression of these two SdmAB homologs from two different strains together taxes *E. coli* in a way that is toxic and ends up effecting lanosterol production.

L94. It would be useful to compare in *E. salina* quantities of steroids per g cell biomass from direct extraction versus from extraction after acid hydrolysis. This would give an indication of the relative abundance of free steroids versus conjugates. Currently, it is unclear if both occur in substantial amounts in the cells.

We have included a supplementary table with the concentrations of cholesterol, desmosterol, zymosterol, and 25-OCH present in both Bligh Dyer and acid hydrolysis extractions and included relevant discussion in the text.

Our quantification of 25-hydroxycholesterol, a product of degradation of sterol conjugates in *E. salina*, would suggest it is present at concentrations within a similar order of magnitude to other sterols present in *E. salina*. If the additional peaks that elute after the 45-minute mark do represent sterol ethers, the concentration of intact sterol conjugate in *E. salina* may be much higher. Identification of the intact sterol conjugate in *E. salina* and quantification of this compound (instead of the degradation products) would provide a more accurate comparison of levels of free and bound sterols in *E. salina* and is a focus of our future research. Additionally, we think we lose sterols through both the sequential lipid extraction process that was used to generate the acid hydrolysis data (lipids were first extracted through Bligh Dyer and purified with Si-column chromatography before treatment with methanolic acid) and degradation from the acid itself. This makes direct comparison of sterol concentration between Bligh Dyer and acid hydrolysis extractions a less biologically meaningful number.

L230. Given the inability to reproduce steroid conjugate synthesis in *Calothrix* sp., it is not appropriate to make comparisons about free versus conjugated steroids in *Calothrix* sp. versus *E. salina*.

In our updated version of the text, we have removed this comparison.

L27. Perhaps add digestion as an important function (bile acids).

This is an interesting function of sterols to consider when thinking about potential functions in the predatory myxobacteria. We included digestion as a function of modified sterols in eukaryotes.

L60. The last paragraph of the Introduction recapitulates the main findings in the Abstract and elsewhere. It would be less redundant to focus here instead on the rationale and approach of the study.

We have changed the last paragraph of the introduction to focus more on why we choose to study *E. salina* and *Calothrix*. Additionally, to fit Nature Communications requirement for abstract length, we have modified the abstract.

L109. This paragraph would be clearer if a supplemental figure showed the cholesterol synthesis pathway, including SMO and OSC, and indicated the names and abbreviations used for the enzymes.

We added SMO, OSC and corresponding substrates to the cholesterol biosynthesis pathways in Fig S2. We also added abbreviations used in the main text to this supplementary pathway figure.

L133 states, "in the presence of lanosterol", which seems to imply exogenous lanosterol was added. It appears that the host strain overproduces lanosterol, so the text would be clearer without the words quoted here.

We removed the quoted text and added clarifying language about the expression system used.

L138. A more accurate conclusion is that SdmAB in *E. salina* are not sufficient to fully demethylate the C-4 position "of lanosterol" (ie, they might do so to another steroid substrate).

That's very well possible. Based on our sterol analysis, we would predict SdmAB to function on 4,4-dimethyl-cholesta-8,24-dienol. We have changed the text as suggested.

Fig S6 does not add much to the manuscript. The statement that steroid synthesis genes are not in clusters in *E. salina* is sufficient without all the gene maps.

One of the questions we have repeatedly received while presenting this work is about the genomic context for sterol biosynthesis genes in *E. salina*. For that reason, we would like to retain this figure in the supplement.

Figs 2 and 4 show the same biosynthetic pathway. To avoid this redundancy, consider a figure that combines the results for both bacterial strains. This would have the added benefit of facilitating comparison of the two.

With our reorganization of the paper, these two pathways now exist as a single supplementary figure which includes both the Bloch and K-R pathway.

L154 should probably refer to Fig. S7.

We have fixed this reference in the main text.

Figs S1 and S8. There is redundancy between these figures. Probably they should be combined. While we recognize that both of these are supplementary figures showing spectra from sterols we identified in the paper, we have split these into two figures to separate the sterols found in *E. salina* and *Calothrix* from the sterol produced by our heterologous expression system.

L123 is a confusing conclusion, because later it is shown that *Calothrix* sp. produces cholesterol, and Table S1 shows that it has homologs for most of the cholesterol synthesis genes in *E. salina*. Perhaps specify in L123 that the conclusion is "based on the stringent search criteria used".

We have added clarifying text to this section.

L217. Conclusion not substantiated. Only one species was conclusively shown to produce cholesterol.

We have changed the discussion to reflect this.

L219. Conclusion not substantiated; what evidence suggests a regulatory role for sterols in bacteria? Should clearly state that this is purely speculative.

We have removed language suggesting a regulatory role for sterols in bacteria.

L260. Change "existence of a potential novel cholesterol biosynthesis pathway" to "possibility of an alternate cholesterol biosynthesis pathway".

With how we have reorganized the paper, discussion of the *Calothrix* pathway was removed.

L267. "These missing homologs" is unclear. No missing homologs were previously discussed.

With how we have reorganized the paper, discussion of the *Calothrix* pathway was removed.

L278 states, "function through a mechanism separate from eukaryotes". Is this correct? Consider that nonhomologous genes may encode proteins with the same biochemical function/reaction mechanism. If reaction mechanisms are not known, it is probably more accurate to discuss how parts of the pathways are not homologous in the various taxonomic groups.

That's a good point. While we know that SdmAB from aerobic methanotrophs remove the beta-methyl group instead of the alpha, representing a mechanistic difference, we have not investigated the stereochemistry driving double demethylation by these enzymes. Based on the oxidation intermediates we see, it's likely that SdmA while being a dioxygenase adds a single oxygen atom to the substrate at a time. Similar functions have been reported for other Rieske-type dioxygenases. We suspect that these SDR-type reductase likely function through the same mechanism. We have changed the language in the discussion to better reflect this, as suggested.

Reviewer #2 (Remarks to the Author):

The authors have done a satisfactory job of responding to review comments. The manuscript is now deemed suitable for publication.